# Growing Cities and Mass Participant Sport Events: Traveling Behaviors and Carbon Dioxide Emissions

**Stavros Triantafyllidis *** and **Harry Davakos**

Department of Health and Human Performance, The Citadel, The Military College of South Carolina,
171 Moultrie St, Charleston, SC 29409, USA
* Correspondence: striantaf@citadel.edu; Tel.: +1-(843)-408-8625

**Abstract:** Fast-growing cities are expected to become a key contributor to the global climate crisis. A key characteristic of those urban districts is the accommodation of mass participant sport events. Sport entities, in collaboration with city governments, plan annually active participation sport events to promote their cities as tourism destinations. Mass participant sport events aim to attract more visitors to the cities and to establish a successful social and economically sustainable future to those places. Given the fact that transportation is a critical factor of the residents and the visitors' behavior, it is crucial to research the quantity of $CO_2$ emissions generated to those places in association with the travel behaviors of the active sport event participants. Data collected from an annual mass participant running event in a highly ranked growing city in the United States. Findings showed that most of the active sport event participants traveled more than 150 miles to participate in the race and they used their vehicles. The largest quantity of $CO_2$ emissions derived from those participants who traveled a round trip of, on average, 500 miles. The long-distance travelers alone generated 338 million kg of $CO_2$ emissions. The conclusions recommend that growing cities and sport events should target long-distance travelers for promotions concerning sustainable transportation. Consequently, mass participant sport events could play a crucial role in the development of growing cities, and, in turn, growing cities that control long-distance traveling behaviors can reduce the global amount of greenhouse gas emissions and their impact on the global environmental destruction.

**Keywords:** travel behavior; sustainable transportation; city development; sport events; carbon emissions

## 1. Introduction

Rapidly developing cities are often associated with an extensive production of greenhouse gas (GHG) emissions [1,2]. Individually, cities with rapid growth in population, infrastructure, and transportation systems are considered "growing cities," and also defined as high-density areas (HDA) [1,2]. According to References [2,3], in an HDA, the population of people living in the urbanized area is relatively high compared to the average number of people that live across other areas in a country or a nation, such as the United States of America [4]. In the literature, there is evidence that residents of HDA in the United States generate the most significant amounts of GHG compared to areas of low density [4]. Specifically, at the urban city level, GHG emissions are associated with environmental pollution and the smog that occurs in large cities [4,5]. Several studies have illustrated that the primary GHG emissions that is significantly associated with the environmental pollution of those cities is the carbon dioxide ($CO_2$) that emitted into the environment by the modes of transportation used by the residents of a city [2–6]. According to the current literature, the mode of transportation responsible for the most substantial quantity of $CO_2$ emissions are the engine motor vehicles owned by residents for their transportation [3,4]. Taken together, the lack of knowledge regarding the ways that people's traveling behavior can change to a more sustainable and the increasingly high amount of $CO_2$

emissions, it can be suggested that extensive research should be conducted—specifically, research on new, innovative strategies regarding growing cities' transportation systems and on residents' choices regarding the mode of transportation they choose for their mobility. Research that provides evidence of residents' traveling behavior could illustrate potential strategic plans for the cities, where the residents would behave in a manner that would reduce the quantity of $CO_2$ emissions existing in the atmosphere of those places. According to References [5,6], growing cities can also act as change agents of the current global environmental crisis. Findings of the References [5,6] have shown evidence of the potential positive contribution of growing cities to the reduction of $CO_2$ emissions if governments and decision-makers plan a strategic, environmentally friendly infrastructure system that can accommodate modes of sustainable transportation [2–5]. Additionally, the development of policies regarding the control of $CO_2$ emissions generated by the residents and the industrial operations of the growing cities will enhance the practical implementation of such effective governmental strategies, which in turn will set the standards for a carbon-neutral future in those cities [3–5].

In the carbon research literature, there is a discourse about the theoretical and the practical implications related to the negative impacts that occur to the natural environment when sport events hosted in HDA, such as the growing cities [5–8]. For instance, some evidence has demonstrated that in growing cities where sport events hosted, the sport event participants alone generate approximately 1 ton of $CO_2$ emissions by their traveling behaviors [3–5]. Additionally, research by Reference [6] found that in the context of active sport event participation, the traveling behaviors together with the sport-related activities (e.g., consumption of sport-related products and services) of the athletes (e.g., active sport event participants) may constitute areas that present a significantly higher quantity of $CO_2$ emissions.

In the literature of sport management, there is evidence that the motivational factors that would influence amateur athletes to participate in running events are highly correlated with their behavioral intentions to offset the carbon footprint that they generate by their sport-related activities if the sport events provide that option [6,7]. For example, running event participant behaviors that generate $CO_2$ emissions include traveling to the place of the sport event and consuming sport-related products, such as bottled water and energy drinks. According to the green mind theory (GMT), people who care for their health and body present an increased potential for care and concern for environmental issues [8]. Given the theoretical aspect of GMT, there is good potential that the theory will apply to people who are involved in sport activities and events, such as the active sport events that take place in growing cities.

Based on the current literature, people who are actively involved and participating in sport events, such as running event participants, present an excellent sample to quantify $CO_2$ emissions generated by their traveling behavior to the growing city that hosts the sport event. Accordingly, the purpose of the current study was to quantify the $CO_2$ emissions generated by the people who traveled to actively participate in a running event that hosted in a growing city. The following research objectives guided the current study:

1. Identify the mileage traveled by the active sport event participants.
2. Determine the modes of transportation used by the active sport event participants.
3. Calculate the $CO_2$ emissions generated by the active sport event participants based on the mileage traveled and the modes of transportation used.

## 2. Sport Events and CO$_2$ Emissions

Several studies have shown that sport events are associated with the generation of considerable amounts of $CO_2$ emissions due to the participants' behaviors, such as traveling-related activities [9,10]. In particular, studies have calculated the $CO_2$ emissions generated by the spectators who attended sport events in the United States [1,9,10]. Findings have shown that spectators may travel long distances with their vehicles to attend a collegiate football game for a day [3,4]. Other studies have focused their attention on mass participant sport events, such as the marathon races, and have demonstrated that a

significantly large number of participants use their vehicles alone or with other people (i.e., carpooling) to participate actively in those sport events [2–4].

## 2.1. Previous Estimation

Research conducted in collegiate football events in the United States calculated the average number of $CO_2$ emissions generated based on the number of spectators the stadium could fit. Specifically, empirical studies have estimated the number of seats that a sport stadium has. Moreover, studies have calculated the quantity of $CO_2$ emissions based on the mileage driven and the mode of transportation used on average by the sport event attendees [1,2]. Specifically, results of previous calculations revealed that each spectator that uses a single-occupancy vehicle (SOV) and drives approximately 80 miles the day of the sport event can generate approximately 15 kg of $CO_2$ emissions. This estimation accounts for sport event attendees' round trip travel distances, e.g., from their home to the stadium and back [1,2]. In support of previous findings, the quantity of $CO_2$ emissions generated by spectators showed that they primarily used SOV (i.e., cars) for round trips (i.e., total mileage driven) within an urban area (less than 80 miles), and that this group of people generates on average between 8 kg and 50.5 kg of $CO_2$ emissions per person [11,12]. The previous estimations illustrate an urgency regarding the environmental degradation occurred by sport event participants' traveling behaviors, which are associated with a significantly large quantity of $CO_2$ emissions.

## 2.2. Provision

People's traveling behavior within growing cities that hosting sport events is a critical problem that can be mitigated if cities plan and apply processes of carbon-neutral urban development [9,10]. In order to achieve a carbon-neutral future, the literature suggests that cities should maintain environmental and social welfare among the residents and their actions in places [9,10]. For instance, policies that would require people to behave responsibly in order to satisfy their needs and wants in their everyday lives may play an important role in the reduction of $CO_2$ emissions in the natural environment [10,11]. Therefore, in growing cities, residents' actions such as transportation and primary usage of SOVs could be managed to mitigate the $CO_2$ emissions generated and the negative effects on the quality of the natural environment. Accordingly, reductions of $CO_2$ emissions may be associated with improvements of the quality of life of residents in growing cities [12,13]. In the literature, several definitions have proposed for sustainable urban development [14,15]. For example, the most common include urban planning regarding a quality growth of infrastructure, and minimization of the depletion of non-renewable resources while considering the needs of future generations [16]. Therefore, the destruction of the natural environment, as well as the degradation of social well-being, is evidence that is warning us to change our behaviors and start building a carbon-neutral future [17–19].

From an urban development standpoint, it can be suggested that cities should utilize carbon-neutral strategies for their urban planning processes by developing infrastructure that can motivate residents to use alternative transportation modes, such as cycling lanes and accessibility to public transportation [13–18]. Additionally, strategies to encourage usage of electric vehicles can be developed and include the availability of charging stations across the cities [1,12–14,16]. Together with evidence from the current literature, the strategies mentioned above can mitigate the usage of SOV significantly, and enhance alternative transportation among the residents of a city [6–9]. Moreover, cities should utilize sport events as platforms for social and environmental interventions, as sport has a positive influence and can motivate people to respect others (e.g., pro-social behaviors), appreciate the outdoor environment, appreciate the natural environment (e.g., pro-environmental behaviors), and consume responsibly for their own health and future generations' wellbeing [1,2,8].

## 2.3. Active Sport Event Participants

In the literature, individuals who attend sport events can be passive sport event participants (e.g., spectators) or active sport event participants (e.g., athletes) [7]. Many studies have focused on

spectators' $CO_2$ emissions and behaviors in sport events [2,7,9]. However, there is a lack of knowledge regarding the amount of $CO_2$ emissions generated by the active sport event participants. Considering the volatile nature of participants involved in sport events (e.g., running races), it is also crucial to explore their traveling behaviors, such as the mileage of travel to the place of the event and their choice regarding the mode of transportation used to attend the sport event. According to GMT, active sport event participants might be a unique case for which to understand more about their habits and behaviors, and, in turn, gain new insights on how the growing cities that host mass participant sport events can become environmentally friendlier and reduce the $CO_2$ emissions generated in the atmosphere.

## 3. Materials and Methods

### 3.1. Study Site

This study took place in an annual mass participant running event in the southeastern United States. The city in which the study took place in Mt. Pleasant in South Carolina. The city that hosted the running event has been identified among the top five growing cities across the United States and is considered the number one most rapid growth city in the southeast of the country [20]. Specifically, the city where the event located is characterized by its rapid urban development [21]. Additionally, the particular place that hosts the running event annually is considered a high-density area with the rapid growth of transportation and infrastructure systems [22].

### 3.2. Participants and Procedures

Web surveys utilized to collect data from the registered runners who participated in the running event that took place on April 6, 2019. The director posted the web survey on the website where participants signed in to check the results of the race. Additionally, one day after the questionnaire first posted, the director of the event sent out a blast email to the registered participants. The web survey sent to 34,924 registered runners for the 2019 running event. The questionnaire built to capture the distance traveled by the participants from their home to the place of the event and back. The respondents indicated the zip code from the place in which they live or reside, and they were asked if they traveled from that location to the event. Participants also asked if they traveled back to that location after the completion of the event, and they were requested to indicate the mode of transportation they used. The structure of the questionnaire assisted the researchers to estimate the distance in miles and the modes of transportation.

### 3.3. Data Analysis

Data screened and cleaned through the Statistical Package for Social Sciences (SPSS) software. After preliminary analysis, N = 929 cases remained for implementation of the research objectives. Descriptive statistics were utilized to explore the mileage groups and the modes of transportation used by the respondents in order to attend the sport event. The descriptive analysis assisted the researchers to investigate the distribution of the data across the six mileage groups (0 to 100 miles, 101 to 200 miles, 201 to 300 miles, 301 to 400 miles, 401 to 500 miles, and 501 or more miles) traveled by the participants. Additionally, groups for the mode of transportation used by the respondents have generated accordingly: (car, carpool, airplane, bus, walk or bike, and others). Based on Reference [23], data screened for possible normality and linearity, possible outliers, and missing values. To implement research objective 3, researchers referred to scientific reports that utilized the greenhouse gases regulated emissions and energy use in transportation (GREET) model to calculate the quantity of $CO_2$ emissions generated by the travelers to the event based on the mode of transportation that they utilized [24,25] The GREET model is a reliable tool that can estimate the $CO_2$ emissions generated by personal automobiles, public transit (e.g., train and bus), and airplanes [26,27]. At the end of the questionnaire, some items captured the demographic characteristics of the respondents.

## 4. Results

### *4.1. Participant Profile*

Results showed that 60.4% of the respondents were female, 96% were more than 41 years old, 69.3% were married, 82.5% had annual household income more than $60,001, 88.4% were white, and 67.7% had completed their bachelor's degree.

### *4.2. Descriptors for Mileage and Mode of Transportation Groups*

In order to implement research objectives 1 and 2, the data analysis reported six different mileage groups and six different modes of transportation groups.

#### 4.2.1. Mileage Groups (Research Objective 1)

Results illustrated that 31.8% of the respondents traveled from 0 to 100 miles, 5.4% of the participants traveled between 101 and 200 miles, 10% traveled between 201 and 300 miles, 5.2% were in the 301 to 400 mileage group, 18.4% traveled between 401 and 500 miles, and finally, 29.3% of the total N = 929 respondents traveled more than 501 miles (see Figure 1). In order to implement research objective 3 and calculate the $CO_2$ emissions generated by the round trips of the participants (i.e., from the departure to arrival point and back) for each mileage group, the average mileage traveled was estimated. A new variable calculated for which the 0 to 100 mileage group was considered to travel 50 miles, the 101 to 200 mile group traveled 150 miles, the 201 to 200 mile group traveled 250 miles, the 301 to 400 mile group traveled 350 miles, the 401 to 500 mile group traveled 450 miles, and 501 or more miles was captured as 500 miles (see Table 1).

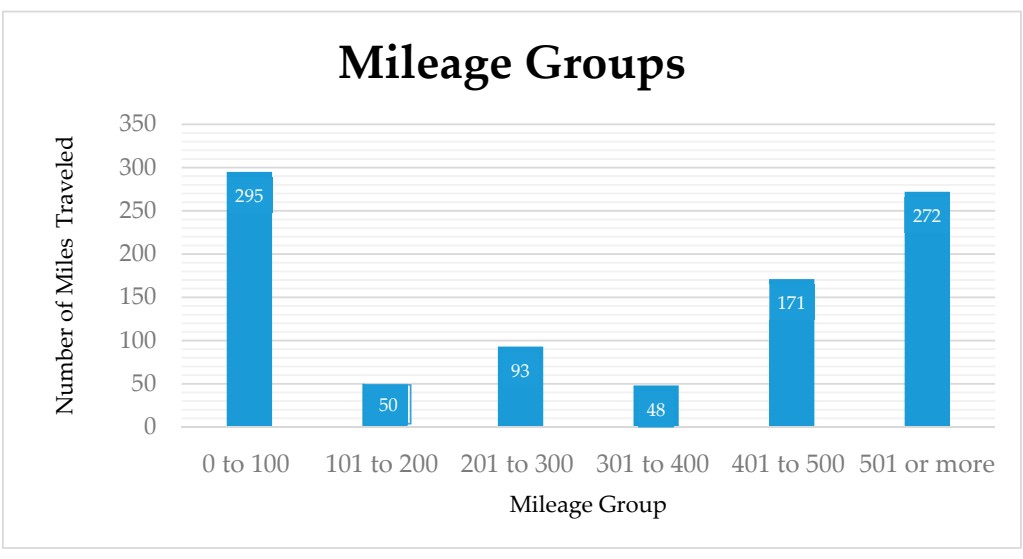

**Figure 1.** Mileage groups for the N = 929 active sport event participants. The columns indicate the number of miles traveled for each group.

**Table 1.** Mileage Groups for the N = 929 Active Sport Event Participants.

| Mileage Groups | Initial Variable | Final Variable [1] |
|:---:|:---:|:---:|
| 1 | 0 to 100 miles | 50 miles |
| 2 | 101 to 200 miles | 150 miles |
| 3 | 201 to 300 miles | 250 miles |
| 4 | 301 to 400 miles | 350 miles |
| 5 | 401 to 500 miles | 450 miles |
| 6 | 501 or more miles | 500 miles |

[1] The final variable illustrates the number of miles used to calculate the $CO_2$ emissions for the participants.

### 4.2.2. Mode of Transportation Groups (Research Objective 2)

The second research objective implemented by developing six modes of transportation groups for the different types of transport utilized by the participants. Results demonstrated that 69% of the participants used their cars and traveled alone, 21.2% carpooled (e.g., used a car with one or more passengers), 4.1% traveled by airplane, 3% by bus (e.g., public transit), 1.8% walked or used their bicycles, and 0.9% indicated that they used other modes of transport without specifying the mode (see Figure 2).

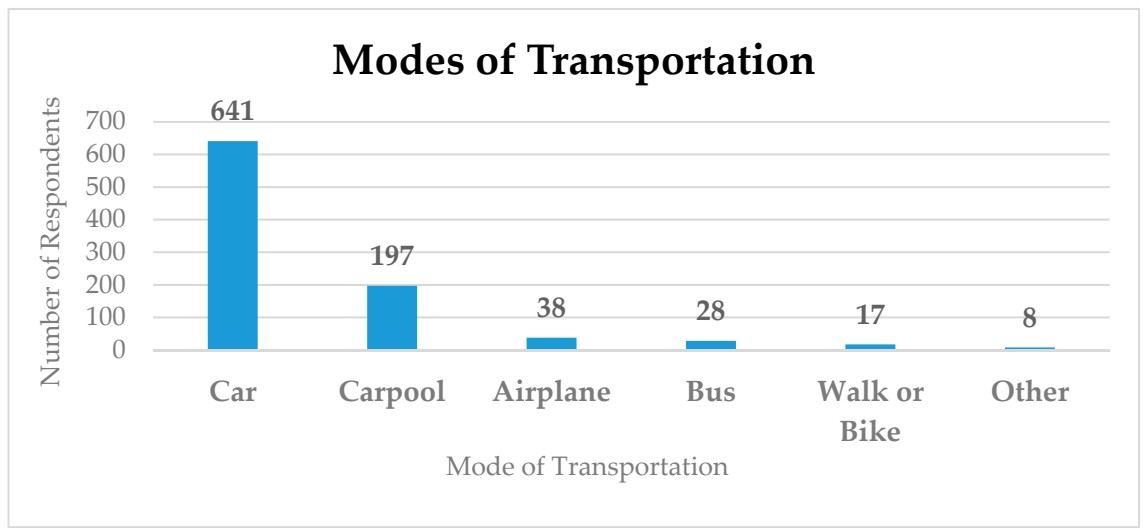

**Figure 2.** Mode of transportation groups for the N = 929 active sport event participants. The columns indicate the number of people that used each transportation mode.

### 4.2.3. Crosstabulation between "500 Miles" Group and "Car Used" Group

In order to estimate the number of people that traveled 500 miles and drove a car alone, a descriptive crosstabulation statistical analysis conducted in SPSS. Based on Figure 2, from the N = 929 responses, n = 641 cases indicated that they used a car, n = 197 carpooled, n = 38 used an airplane, n = 28 used the bus, n = 17 walked or bicycled, and n = 8 used another mode of transportation. Additionally, from the n = 641 who traveled by car and alone (i.e., SOV), n = 176 traveled more than 500 miles, n = 53 carpooled, n = 38 used airplanes, n = 4 used the bus, and n = 1 walked or bicycled. Accordingly, the group of people that traveled 500 miles or more by car and alone represented 18.9% of our sample population (N = 929). Results illustrated that from the actual 34,924 participants, 18.9% represented 6600 actual runners. Thus, 6600 runners traveled alone with their cars more than 500 miles. This number assisted the researchers to calculate the actual $CO_2$ emissions generated by this group of participants in the running event that took place on 6 April 2019.

### 4.3. $CO_2$ Emissions (Research Objective 3)

The purpose of research objective 3 was to measure the $CO_2$ emissions generated by the active sport event participants based on the mileage traveled and the mode of transportation used. Given the purpose of the study and the literature that highlights the critical aspect of the quantity of $CO_2$ emissions generated by active sport event participants, and the results of research objective 2, research objective 3 was calculated using only the respondents who traveled alone in their cars more than 500 miles (see Table 2) [1–3].

According to the United States Environmental Protection Agency (US EPA) reports, on average, the typical American car generates an average of 8.9 kg of $CO_2$ emissions per gallon of regular gasoline [22]. Given the amount that Reference [22] has provided, the average American car generates 0.40 kg of

$CO_2$ emissions per mile, and therefore, based on Reference [23], the carbon footprint of one person inside a given vehicle driving 1 mile was assumed to have a carbon footprint of 0.40.

**Table 2.** Groups of $CO_2$ Emissions Generated per Mileage Driven Group for Car Mode.

| $CO_2$ Emission by Transportation | Average Mileage Driven Group | Mode of Transportation [1] |
|:---:|:---:|:---:|
| 1 | 50 miles | |
| 2 | 150 miles | |
| 3 | 250 miles | Car |
| 4 | 350 miles | |
| 5 | 450 miles | |
| 6 | 500 miles | |

[1] The total number of respondents who drove a car was n = 641.

### 4.3.1. Mathematical Formula for $CO_2$ Emissions per Person

For calculation of the $CO_2$ emissions, the average mileage driven by the participants multiplied by 0.40 kg of $CO_2$ emissions. The following equation used:

$CO_2$ emissions by car per person = (0.40 kg of $CO_2$ emissions) × (average mileage driven per group)

Table 3 shows the quantity of the $CO_2$ emissions generated by participants who traveled by car for each mileage group.

**Table 3.** The Quantity of $CO_2$ Emissions Generated per Person and Total per Mileage Group that Drove a Car Alone.

| Groups for $CO_2$ Emission by Car for Each Mileage Group | Quantity of $CO_2$ Emissions Per Person | Number of People Per Mileage Group [1] |
|:---:|:---:|:---:|
| 1 | 20 kg | 175 |
| 2 | 60 kg | 41 |
| 3 | 100 kg | 79 |
| 4 | 140 kg | 41 |
| 5 | 180 kg | 129 |
| 6 | 200 kg | 176 |

[1] The amount of $CO_2$ emissions generated by one person for each given group who drove a car for n = 641.

### 4.3.2. Mathematical Formula for N = 641 $CO_2$ Emissions per Mileage "Car Used" Groups

Total $CO_2$ emissions by mileage group = (kg of $CO_2$ emissions per mileage group) × (number of participants per mileage group)

### 4.3.3. Mathematical Formula for $CO_2$ Emissions per Person per Mileage Driven per with a Total 34,924 Participants

Total $CO_2$ emissions by mileage group = (kg of $CO_2$ emissions in total per 500 miles group per car mode = 35,200 kg of $CO_2$ emissions) × (actual participants = 6600) (divided)

(500 Miles Group drove Car Alone of Sample = 176)

The results showed that 6600 actual active sport event participants generated 1,320,000 kg of $CO_2$ emissions just from their round-trip travel for the sport event.

## 5. Discussion

This research investigated the traveling behaviors of active sport event participants in order to estimate the total amount of $CO_2$ emissions generated when a mass participant sport event hosted in a growing city. The findings illustrated that the most significant amount of $CO_2$ emissions derived from those participants who traveled more than 500 miles in round trip distance from their initial location. As those participants indicated in their responses, they mostly departed from their homes to actively participate in the sport event, and they traveled back from the place of the sport event (i.e., the growing city) to their homes. Additionally, the most substantial amounts of $CO_2$ emissions generated by those participants identified as individuals who traveled more than 500 miles with their cars and alone. These active sport event participants mainly drove their vehicles alone without any other passengers. According to the literature, these people can be identified as using an SOV [11]. Consequently, the results of this study constitute valuable information for the governments and decision-makers of growing cities in the United States. Specifically, policy development and strategic plans that could mitigate $CO_2$ emissions will be a crucial step forward towards carbon-neutrality. Growing cities have a role as change agents and can act as role models for other places to act responsibly and set strategic plans that accommodate sustainable transportation. Those practices can be the initiatives for changes in other industries beyond transportation, where $CO_2$ is the primary outcome of their operations.

### 5.1. Managerial Implications

This study has contributed to the literature of carbon research, as there is a need to understand the carbon footprint that is left behind when sport events are hosted [1–3]. Specifically, the findings provide crucial information regarding the $CO_2$ emissions generated from growing cities when a mass participant sport event is hosted [4–6]. Accordingly, an enormous impact comes from those sport event participants that choose to drive their cars alone, and that live approximately 500 miles or more from the destination of the sport event. Findings also illustrated that sport events tend to attract people that live far away from the places of the events. This outcome can be interpreted in two ways. First, there is an increasing social and economic impact on the host city, as most of the sport consumers (i.e., registered active participants) are considered sport tourists, and they spend time and money in the host places. In turn, this is associated with a negative environmental impact, as their traveling behaviors contribute to a vast quantity of $CO_2$ emissions [6,7].

Consequently, there is compelling evidence that cities hosting sport events and sport entities should control the traveling behaviors of this large group of participants. According to Reference [4], new strategic planning has to be developed by the sport organizations that host mass participant sport events in order to accommodate and attract people (i.e., positive social and economic impact), but facilitate environmentally friendlier transportation modes. For example, according to Reference [3] and in support of the strategic recommendations by Reference [4], sport event organizations should include motivating long-distance travelers to carpool with other active sport event participants. This tactic can include incentives to those traveler participants, such as discounts on their registration fee. For example, if SOVs carpool with at least one passenger, the quantity of $CO_2$ emissions generated will be decreased by half; if there are two passengers, then each participant will generate one-third of what would be generated by only one person traveling in the vehicle (i.e., SOV).

### 5.2. Theoretical Implications

The current study provided some insights into GMT [8]. Specifically, GMT indicates that active and healthy people have care and concern for environmental quality, and therefore, for the carbon footprint they leave behind. This theory implies that individuals who respect themselves and take care of their health and body will be more likely to care for and protect the natural environment, as well as other individuals [7].

Recent studies [4–8] have also suggested that urban development practices regarding transportation systems should be implemented by governments of growing cities and sport organizations that host mass participant sport events, in order to foster a new trend towards the usage of sustainable transportation. Ultimately, an appropriate decision-making process by leaders of cities and sport organizations can allow people to choose alternative modes of transportation that generate the least $CO_2$ emissions, and support the development of innovative transportation systems that encourage alternative modes of transportation. Accordingly, based on the findings of this study and supported by GMT and strategically responsible natural environment management, residents of the growing cities across the globe and sport event participants' traveling decisions can become more environmentally conscious. Individuals involved in sport are more likely to make healthier choices regarding impacts on their health and the environment [4,8]. In conclusion, this research suggests that significant mass participant sport events should be held in cities where there is an excellent public transportation system with a low environmental burden.

### *5.3. Limitations and Future Research*

The current research estimated $CO_2$ emissions from participants who traveled more than 500 miles alone, using their vehicles. However, there was evidence that a large number of people traveled within a radius of 0 to 100 miles using their cars and alone. Those participants can be considered residents of the growing city that hosted the mass participant sport event, and there is evidence that they contribute a large amount of $CO_2$ emissions by their transportation choices, i.e., cars. Indeed, the most considerable amount generated by the long-distance traveling participants, but future research should focus on the carbon footprint of the residents, who may demonstrate a broader spectrum of sustainable transportation choices.

**Author Contributions:** S.T. and H.D. conducted this research.

**Funding:** This research received no external funding.

**Acknowledgments:** This study was conducted successfully with the crucial help and support of the Cooper River Bridge Run (CRBR) organization. We appreciate the CRBR directors that helped us with the distribution of the web-surveys and their professional collaboration and communication before and after the CRBR event 2019.

**Conflicts of Interest:** The authors declare no conflict of interest.

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
