# Peer review of "Growing Cities and Mass Participant Sport Events: Traveling Behaviors and Carbon Dioxide Emissions"

_carbon, 2019_

Round 1

Reviewer 1 Report

An interesting topic and relevant to various sports events across the world. However a very small sample presented (one event only) and there is no discussion of how this paper could impact the vast number of such sport events globally. What is the link with sustainable urban development mentioned in the beginning? 

Some areas discussed in the detailed PDF comments document need to be more powerful in order to make the overall paper stronger. The impact of such a measurement method and suggestions of how it can be used would be necessary to enhance the discussion. 

Please see detailed comments document. 

Author Response

Thank you very much for your support and crucial comments. Please find attached our responses.

Reviewer 2 Report

l  I think the previous part of section 2.1 is too long. I recommend the author that this part (L.80 ~L.132) should divide two parts. For example, the first part (L.80 ~L.110) is named 2.1 Previous Estimation, and second part (L.111 ~L.132) is named 2.1 Provision. If the author would accept these changes, section No.2.1 should change 2.3.

l  There are two case arcs in line 130. Please correct right form.

l  In lines of 137 and 138, there are two symbols of [cite]. If this symbol mean cite literature, please correct right form.

l  In L.144, there is a word "hist". I can't understand the meaning of this word. Is this word "hist.”?

l  The author required the web-survey questionnaires such as traveled distances, zip code and mode of transportation, etc. to the participants through the director one hour before the start of the running race. I think the participant's attributions like those are easy to check the previous leistered information. Why did the author do these procedure, please write more detail in section 3.2.

l  Although I tried to check N=176 in L.240 which is very important number at later section, I couldn't understand how to calculate. Please explain much detail how to calculate this number.    

l  In L.279, is this sentence continue to next page? please correct right form.

l  In L.356, it may be necessary to apostrophe in the back of "transportation". Please check.

l  As to section 5.2, I think the author should suggest that the big sports event should be held at the city where have good public transportation system with low environmental burden.

Author Response

Thank you for your kind words and crucial review. Please find attached our responses.

Round 2

Reviewer 1 Report

The authors made an effort to address the comments, however there are still some areas where the paper could be stronger. 

Introduction: Literature was added, but it stil reads as two separate cases, where you analyse the growing and green cities in the beginning and then mentioning the sport events at the end. The introduction should give the critical literature but should also explain to the reader the basis of your research, briefly the reason behind it and the impact. 

Materials and methods. The specific method is described, however there is no response to my comment for one case study only. It can be a pilot study or a first experiment, but then an explanation should put in justifying why one case study only has important findings to be taken under consideration. Plans for other similar studies to cross check findings would make more scientific readers to feel secure. 

Discussion. It has been improved, but it still reads vague in some parts. You will need to extract the results from the previous section and give them to your readers in a simple and compact way explaining why your research is so important. What is significant? 
By mentioning that urban centres and cities are able for change you do not address why is this relevant to sport events? There are many papers on carbon neutral urban design why is this relevant to your case? 
Also, literature can be mentioned in discussion, but it is mostly about discussing your own results and suggesting what should be done. What are your suggestions based in your reseach and why is this strong enought to be adopted by other cities? 

Perhaps it would be easier if you add a conclusion section where you mention your overall conclusions and make your suggestions. 

Author Response

Please find attached our detailed comments. We really appreciate your help and support.

Please let us know if you have any other questions.

Reviewer 2 Report

Why the author didn't correct the items that I pointed out at previous review?

Author Response

Thank you for your kind words and critical comments. Please find our detailed response to the file attached. Thank you.

Round 3

Reviewer 1 Report

There is significant improvement to the article. Please find some minor corrections below. 

Please rewrite the number of CO2 to subfix in line 44

Line 72  ''consumption s'' remove 's'

Line 73  'possibly present' instead of 'may' ? 

Line 115 'research conducted' remove 'that' 

Line 118 ' a sport stadium has'

line 146, 'exist' . Does this belong there? 

Line 161, is it urban or strategic? Usually these are two different scales. 

Line 438 ' this research suggests' and either 'a big mass-participation sport event' or 'big mass-participation sport events' 

Author Response

Dear reviewer:

Please find below our responses to your comments:

Please rewrite the number of CO2 to subfix in line 44

Authors: CO2 is subfixed.

Line 72  ''consumption s'' remove 's'

Authors: Thank you, 's' removed.

Line 73  'possibly present' instead of 'may' ? 

Authors:

Line 115 'research conducted' remove 'that' 

Authors: 'that' removed.

Line 118 ' a sport stadium has'

Authors: Have was replaced by has.

line 146, 'exist' . Does this belong there? 

Authors: Exist removed.

Line 161, is it urban or strategic? Usually these are two different scales. 

Authors: Thank you. Strategic removed. We refer to the urban planning process.

Line 438 ' this research suggests' and either 'a big mass-participation sport event' or 'big mass-participation sport events' 

Authors: We rephrased it: this research suggests that big mass-participation sport events 

Reviewer 2 Report

I checked your re-revised manuscript based on my comments.

All the items excluding the following were corrected. Please check.

Does line 341 of the updated manuscript continue?

There are two "4.3.1" in lines of 329 and 357 of your updated manuscript.

Please rewrite the number of CO2 to subfix in line 368 of your updated manuscript.

Author Response

Dear reviewer:

Please find below our responses to your comments:

Does line 341 of the updated manuscript continue?

Authors: Thank you. We checked the sentence and it is concluded on the aforementioned line.

There are two "4.3.1" in lines of 329 and 357 of your updated manuscript.

Authors: Thank you for your notice. The second subtitle updated as 4.3.2.

Please rewrite the number of CO2 to subfix in line 368 of your updated manuscript.

Authors: CO2 was subfixed in the entire manuscript.